# One-Year Outcome of Combination Therapy with Full or Reduced Photodynamic Therapy and One Anti-Vascular Endothelial Growth Factor in Pachychoroid Neovasculopathy

**DOI:** 10.3390/ph15040483

**Published:** 2022-04-15

**Authors:** Miki Sato-Akushichi, Shinji Ono, Tatsuro Taneda, Gerd Klose, Asuka Sasamori, Youngseok Song

**Affiliations:** 1Department of Ophthalmology, Asahikawa Medical University, Asahikawa 078-8510, Japan; shinji.ohno@gmail.com (S.O.); kyokui130090@asahikawa-med.ac.jp (T.T.); ysong@asahikawa-med.ac.jp (Y.S.); 2Sapporo Ohno Eye Clinic, Sapporo 065-0024, Japan; 3Carl Zeiss Meditec, Inc., Dublin, CA 94568, USA; gerd.klose@zeiss.com; 4Nayoro City General Hospital, Nayoro 096-0017, Japan; s-asuka@asahikawa-med.ac.jp

**Keywords:** anti-vascular endothelial growth factor, photodynamic therapy, combination therapy, pachychoroid neovasculopathy

## Abstract

This paper evaluates a one-year treatment outcome after full or reduced photodynamic therapy (PDT) and anti-vascular endothelial growth factor (VEGF) combination therapy for pachychoroid neovasculopathy (PNV). After the initial combination therapy, a total of 29 eyes from 29 patients (16 for full treatment and 13 for reduced treatment), exhibited reduced, central retinal thickness and central choroidal thickness, and the improvements were maintained for 1 year after the initial combination therapy. Twenty-two eyes (75.9%) required no additional treatments for 1 year. The recurrence rate was 31.3% in the full treatment and 15.4% in the reduced treatment, with no significant differences between them. One shot of anti-VEGF and full or reduced PDT combination therapy had similar efficacy in treating PNV. Further prospective, large-scale, and long-term studies are required to determine a better treatment for PNV.

## 1. Introduction

Pachychoroid spectrum disease encompasses a disease spectrum characterized by functional and structural abnormalities of the choroid, including thick choroid, dilated choroidal vessels with inner choroidal attenuation, choroidal vascular hyperpermeability, intravortex venous anastomosis, and asymmetry of watershed zone [1,2,3]. The spectrum includes central serous chorioretinopathy, pachychoroid pigment epitheliopathy, pachychoroid neovasculopathy (PNV), polypoidal choroidal vasculopathy (PCV), focal choroidal excavation, and peripapillary pachychoroid syndrome [4,5]. Specifically, PNV is different from neovascular age-related macular degeneration (AMD) both phenotypically and genetically [6]. Furthermore, some reports consider PCV as PNV with polypoidal lesions and PNV as PNV without polypoidal lesions [7,8]. 

The EVEREST II study has reported that PCV combination therapy using ranibizumab and photodynamic therapy (PDT) showed more favorable visual outcomes and higher rates of complete polyp regression and required fewer intravitreal injections than the anti-vascular endothelial growth factor (VEGF) agent of ranibizumab monotherapy [9]. Specifically, the presence of grape-like polyp clusters and large lesion-size polyps presents a high risk of reactivation of PCV [10,11,12]. PCV with thicker subfoveal choroidal thickness, choroidal vascular hyperpermeability, and the grape-like polyps suggest that the combination therapy of PDT with anti-VEGF agents may reduce the treatment burden [13,14]. PNV is a favorable outcome of anti-VEGF therapy, with a longer retreatment-free interval than typical neovascular AMD after initial loading injections [15,16]. Moreover, adjunctive PDT treatment in eyes with PNV refractory to anti-VEGF monotherapy resulted in complete fluid absorption in most eyes and visual improvement within 1 year [17]. Moreover, recent studies have demonstrated the efficacy of the combination therapy of PDT and the anti-VEGF injection for PNV [2,7,18]. However, treatment protocols differ from each report, both in the total dose or fluence of PDT, and in the total number of anti-VEGF. Some reports conducted full- or half-fluence- or half-dose PDT, combined with one or three consecutive anti-VEGF. Thus, a standard-treatment protocol has not been established [7,8,18,19]. Additionally, there are no reports comparing full-PDT combination therapy with half-PDT combination therapy. The purpose of this present study is to investigate the 1-year outcome of one anti-VEGF and the full- or the reduced-PDT combination therapy for PNV.

## 2. Results

A total of 29 patients with PNV diagnoses (25 men and 4 women) were enrolled. The mean age of the subjects was 68.3 (standard deviation; 8.5) years; the mean logarithm of the minimum angle of resolution (logMAR) was 0.19 (0.29, range: −0.08 to 1); the mean central retinal thickness (CRT) was 305.2 (104.8) μm; and the mean central choroidal thickness (CCT) was 374.4 (73.2) μm. Nine patients (31.0%) had past treatment history, and based on physician-diagnosed past history and medication use, 3 (10.3%) had diabetes mellitus and 13 (44.8%) had hypertension. Nineteen patients (65.5%) had a smoking history, while three (10.3%) had no smoking history information. 

The patients were divided into treatment conditions: 16 patients with full-dose PDT and 13 patients with reduced PDT (11 patients treated with the half-fluence regimen and 2 treated with the half-dose regimen). During anti-VEGF therapy, in the full-treatment group, aflibercept and ranibizumab were given to 2 patients and 14 patients, respectively, and in the reduced-treatment group, aflibercept and ranibizumab were given to 3 patients and 10 patients, respectively. Seven (43.8%) patients in the full-treatment group and two (13.3%) patients in the reduced-treatment group had past treatment history. One (6.3%) patient in the full-treatment group and two (15.4%) patients in the reduced-treatment group had diabetes mellitus, and seven (43.8%) patients in the full-treatment group and six (46.2%) patients in reduced-treatment group had hypertension. Furthermore, 12 (75.0%) patients in the full-treatment group and 7 (53.8%) patients in the reduced-treatment group had smoking history, although there were no smoking history information for 2 people (12.5%) in the full-treatment group and 1 (7.7%) person in reduced group. The demographics of the patients at baseline are shown in Table 1. There was no statistically significant difference between these two treatment groups. 

The mean logMAR at 1, 3, 6, and 12 months after the combination therapy was 0.21 (0.34), 0.17 (0.28), 0.16 (0.30), and 0.16 (0.30), respectively. The mean CRT at 1, 3, 6, and 12 months after the combination therapy was 190.1 (76.9), 177.9 (78.9), 179.6 (84.6), and 174.1 (86.5), respectively, and the mean CCT at 1, 3, 6, and 12 months after the combination therapy was 284.5 (80.2), 298.8 (87.5), 293.3 (92.0), and 294.3 (95.3), respectively. Among logMAR, there were no significant differences from baseline to 1 year after the treatment (*p* value > 0.05 for each comparison). On the other hand, significant reductions were seen in CRT and CCT 1 month after the combination therapy compared to the baseline, and the reductions in CRT and CCT were maintained 1 year after the treatment. 

Divided by treatment condition, each treatment group showed no significant differences in logMAR from baseline to 1 year after the treatment (*p* value > 0.05 for each comparison). However, significant reductions were seen in CRT and CCT 1 month after the full and reduced treatments compared to the baseline, and the reductions in CRT and CCT were maintained 1 year after the treatment. Compared between treatment groups, no significant differences were found in logMAR, CRT, and CCT at any time point (*p* value > 0.05 for each comparison). The serial changes in the mean logMAR, CRT, and CCT during the 1-year follow-up after the combination therapy, divided into the full or reduced treatments, are shown in Figure 1 and Figure 2.

After the initial combination therapy, 22 eyes (75.9%) required no additional treatments for 1 year, and among them, 6 eyes had past treatment history such as anti-VEGF monotherapy, PDT monotherapy, and photocoagulation. Among the patients with past treatment history, no dry macula or recurrence of fluids were found in 1 person after one month, in 1 person after 6 months, and in 1 person after 9 months within the 12 months of follow-up exams. Divided by treatment condition, 11 eyes (68.8%) in the full-treatment group and 11 eyes (84.6%) in the reduced-treatment group required no additional treatment. Divided into the groups with or without past treatment history, 6 out of 9 people (66.7%) and 16 out of 20 people (80.0%) required no additional treatment. Table 2 and Table 3 show the recurrence rate and the total number of additional anti-VEGF treatments during the 1-year follow-up.

### 2.1. Case Presentations

#### Case 1: A 58-Year-Old Male (Reduced Treatment, no Recurrence)

A 58-year-old male with recurrent subretinal fluid in his left eye was referred to our hospital. He was previously diagnosed with central serous chorioretinopathy, but his symptoms improved without any treatment. Subretinal fluid relapsed 14 months later, and LogMAR in his right eye was 0.398. Subretinal fluid and the retinal pigment epithelium (RPE) abnormality were seen in fundus photography and optical coherence tomography (OCT). Choroidal neovascularization was seen in OCT angiography and the fluorescein angiography (FA) showed a faint leak, and the indocyanine green angiography (ICGA) showed hyperfluorescein spots, dilated choroidal vessels, and choroidal vascular hyperpermeability. We diagnosed him as having PNV. After the half-fluence PDT (spot size 4600 μm) and ranibizumab were applied, the subretinal fluid disappeared completely until the one-year follow-up. Visual acuity improved to 0.222 LogMAR (Figure 3).

#### Case 2: A 72-Year-Old Male (Full Treatment, Recurrent at Five Months after the Treatment)

A 72-year-old male with a finding of macular abnormalities in his right eye was referred to our hospital. LogMAR was 0.398. Subretinal fluid, RPE abnormality, dilated choroidal vessels, thick choroid with inner choroidal attenuation were seen in the OCT. FA showed a diffuse leakage, and ICGA showed multiple hyperfluorescein spots, dilated choroidal vessels, and choroidal vascular hyperpermeability. We diagnosed him as having PNV. Full-dose PDT (spot size 6000 μm) and ranibizumab were administered. Subretinal fluid disappeared 1 month after the treatment, but it recurred 5 months after the treatment. At the point of recurrence, an anti-VEGF injection (aflibercept) was administered. Although the one dose of anti-VEGF was effective, a new polyp was observed at one-year post-treatment (Figure 4).

## 3. Discussion

In this study, we demonstrated the efficacy of the combination therapy of the full treatment (full-PDT and one anti-VEGF agent) and the reduced treatment (half-dose or half-fluence PDT and one anti-VEGF agent). Both treatments improved the subretinal fluid and decreased CCT, and the reductions in CRT and CCT were maintained 1 year after the treatment. 

PNV occurs due to a pachychoroid-driven process involving choroidal congestion, and choroidal vascular hyperpermeability manifested by choroidal thickening and dilated choroidal vessels [20,21]. Furthermore, PNV can develop into PCV [22]. The PDT with verteporfin affects choroid and causes choriocapillaris narrowing, choroidal hypoperfusion, reduction in choroidal exudation, and choroidal vascular remodeling. These phenomena include normalization of the dilated and congested choroidal vasculature [23,24,25]. However, choriocapillaris ischemia, secondary choroidal neovascularization, transient impairment in retinal function, and RPE atrophy are common side effects of the PDT [23,26,27]. Although most subretinal hemorrhage absorbs without treatment, post-PDT hemorrhage can occur occasionally and lead to a vitreous hemorrhage and poor visual prognosis [28]. Previous studies have suggested that the combination therapy with PDT and an anti-VEGF agent have some advantages over PDT monotherapy, such as improving vision, reducing the recurrence of polyps, reducing fluid leakage and inflammation, and fewer hemorrhagic events [29,30]. Moreover, the combination therapy prevents secondary choroidal neovascularization formation, modulating VEGF upregulation from PDT-induced tissue hypoxia [31]. Although a half dose or fluence of PDT has been evaluated in PNV treatment, treatment protocols differ from each report, and a standard treatment protocol has not been established yet. We compared the treatment outcome of one anti-VEGF and the full or the reduced PDT. In our results, there were no statistically significant differences in the recovery of logMAR, reduction of CRT, reduction in CCT, and the recurrence rate between the two treatment protocols. Both combination therapies have obtained favorable outcomes in treating PNV. 

Previous studies reported recurrence rates of 23.8% [7] and 45.5% [18] in the full-dose PDT and one or three anti-VEGF agents, and 19% [9] and 53% [32] in the half-dose or the half-fluence PDT and one anti-VEGF agents. In our result, the recurrence rate was 24.1% in total, 31.2% in the full treatment and 15.4% in the reduced treatment, which are in agreement with the previous studies. However, the recurrence rate was lower in the reduced-treatment group than in the full-treatment group, despite no statistically significant differences in the recurrence rate in this study. In the full-treatment group, seven eyes with past treatment history had a higher recurrence rate than those (nine eyes) with no treatment history. Moreover, eyes with and without a previous treatment history exhibited 1-year post-treatment recurrence rates of 33.3% and 20.0% in total, 42.9% and 22.2% in the full treatment, and 0% and 18.2% in the reduced treatment, respectively. It is difficult to compare the recurrence rates due to the small sample size, especially with the previous treatment history in the reduced group (*n* = 2). However, there was a trend that the recurrence rate was higher in eyes with a past treatment history than those with no previous treatment history. Hence, the recurrence rate was higher in the full-treatment group in this study. 

At 3 and 6 months, 88.9% (eight of nine people) and 77.8% (seven of nine people) of the patients with past treatment history, respectively, showed dry macula, while at 12 months, 66.7% (six of nine people) showed no recurrence. Among the people who had previous treatments, 66.7% (six of nine people) required more than five anti-VEGF agents before achieving dry macula. A previous study reported that adjunctive PDT treatment of PNV refractory eyes with anti-VEGF monotherapy resulted in 86% of dry macula in 3 months and 61% prevention of recurrent exudation for 12 months after treatment [17]. Thus, we can infer that the combination therapy was also effective for anti-VEGF monotherapy-resistant PNV. 

There were several limitations in this study. First, as a retrospective study, a patient selection bias could have played a role. Second, different OCT systems were used for measuring CRT and CCT. In addition, CRT and CCT were calculated manually. Measuring choroid thickness is not easy because the chorioscleral interface is a transient zone, the definition of posterior boundary remains inconstant, and the thick choroid attenuates the OCT signal [33]. Third, two different anti-VEGF agents were used for intravitreal injections, and the choice of anti-VEGF agents and retreatment decision was left to the physician’s discretion. Finally, half-dosed and half-fluence PDT were mixed in the reduced-treatment group. Future studies with a larger sample size are required in order to evaluate the appropriate treatment protocol for eyes with PNV, and other confounding factors should be eliminated.

## 4. Materials and Methods

### 4.1. Population

Patients with a diagnosis of PNV at Asahikawa Medical University Hospital from April 2019 to January 2022, were enrolled. PNV was diagnosed when all of the following criteria were met. (1) Macular neovascularization (MNV) was diagnosed by OCT angiography or FA/ICGA. (2) A shallow irregular RPE detachment at the site of MNV, observed on OCT B-mode images and network vessels of MNV, was detected between the detached RPE and Bruch’s membrane on OCT angiography. (3) Clinical and anatomical features of the pachychoroid phenotype were present, i.e., pathologically dilated outer choroidal vessels (pachyvessels) and attenuation of choriocapillaris on OCT images, regional choroidal vascular hyperpermeability on ICGA images, and anastomosis or asymmetry of choroidal vessels on en face OCT. Eyes with initial development of central serous chorioretinopathy and characteristics serous retinal fluid and/or sub RPE fluid followed by MNV were included; four eyes met such criteria in this study. Patients of this study underwent comprehensive eye examinations, including best-corrected visual acuity (BCVA), slit-lamp biomicroscopy for anterior and posterior segment examination, fundus photography, FA and ICGA, OCT (RTVue XR; Optovue, Inc, Fremont, CA, USA, and Spectralis HRA + OCT; Heidelberg Engineering, Heidelberg, Germany), and OCT angiography (RTVue XR Avanti with AngioVue; Optovue, Inc, Fremont, CA, USA, and PLEX Elite 9000; Carl Zeiss Meditec, Dublin, CA, USA). CRT was manually measured from the inner neurosensory retina surface to the inner of RPE at the foveal center. CCT was also manually measured from the outer surface of RPE to the inner surface of the sclera at the foveal center, and 12 mm × 12 mm OCT angiography images were taken with the PLEX Elite 9000 swept-source OCT and analyzed on the Advanced Retina Imaging Network platform, providing a fully automated choroid quantification algorithm developed by ZEISS. After the analysis, a choroidal thickness map after the automatic segmentation of the choroidal edge was obtained [34]. BCVA was converted to logMAR and used for statistical analysis. OCT scans and OCT angiography images were obtained at the baseline and 1 month, 3 months, 6 months, and 12 months after the combination therapy. The exclusion criteria included (1) the presence of other ocular diseases including diabetic retinopathy, retinal vein occlusion, uveitis, myopic choroidal neovascularization; (2) any history of intraocular surgery within the past six months; (3) low-quality OCT angiography images because of media opacities, poor corporation with taking images; and (4) MNVs with polyps.

### 4.2. Treatment Conditions

All eyes received one intravitreal injection of anti-VEGF agents including 0.5 mg ranibizumab (Lucentis; Novartis Pharma AG, Basel, Switzerland, and Genentech, South San Francisco, CA, USA) or 2.0 mg aflibercept (Eylea; Regeneron, Tarrytown, NY, USA) combined with PDT. PDT was performed on the same day or within 1 week of the intravitreal injection. Intravenous verteporfin was administered over a period of ten minutes. A diode laser emitting at 689 nm coupled into a slit lamp system (Coherent Inc., Palo Alto, CA, USA) was used at an irradiance of 600 mW/cm^2^ over 83 s starting 15 min after the start of the infusion. The treatment spot was determined using the diameter of the largest circle that covered the area of leakage by FA or the area of hyperpermeability by ICGA or both. An additional 1000 μm margin was added to ensure complete coverage of the lesion during light exposure. The choice of half of the normal dose or half of the normal fluence was determined by the provider performing the treatment (S.O.). The total light energy was set at 25 J/cm^2^ for the half-fluence approach, and the total verteporfin dose was set at 3 mg/m^2^ body for the half-dose approach. We added an anti-VEGF injection if residual subfoveal fluid, intraretinal fluid accumulation, or new onset hemorrhage were seen after 1 month of combination therapy. The recurrence period was defined when the accumulation of fluids occurred again, and if recurrent subfoveal fluid or intraretinal fluid accumulated and hemorrhage were observed, anti-VEGF injection was added.

### 4.3. Statistical Analysis

Statistical analysis was performed using the free software R version 4.0.5 (The R Foundation for Statistical Computing Platform, Vienna, Australia) and EZR (Jichi Medical University, Saitama, Japan), a graphical user interface for R [35]. Mann–Whitney U test and Fisher exact test were used to compare the full-treatment group and the reduced-treatment group. The Wilcoxon signed-rank test was used for changes in logMAR, CRT, and CCT in each sector after the treatment, and the Bonferroni methods’ was used to adjust the *p* values. Data are presented by mean (standard deviation), and a *p* value of <0.05 was considered statistically significant.

## 5. Conclusions

This study examined a one-year outcome of anti-VEGF and full- or reduced-PDT combination therapy. Both treatment protocols were similarly effective for PNV anatomically and exhibited a lower recurrence rate, even with previous treatment history. Further prospective, large scale, and long-term studies are required to determine the best treatment protocol for PNV.

## Figures and Tables

**Figure 1 pharmaceuticals-15-00483-f001:**
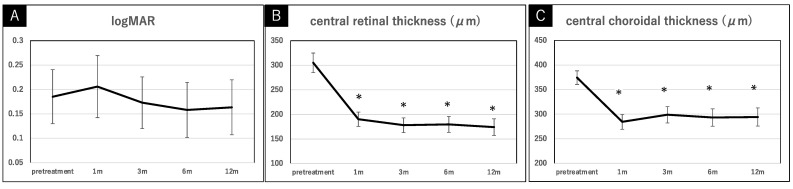
The serial changes in the logMAR (**A**), central retinal thickness (**B**), and central choroidal thickness (**C**) mean during the 1-year follow-up after the combination therapy in pachychoroid neovasculopathy. After treatment, there were statistical differences in the central retinal thickness and the central choroidal thickness at each time period, compared to before treatment. Data are expressed as mean and standard error bar and * *p* < 0.01.

**Figure 2 pharmaceuticals-15-00483-f002:**
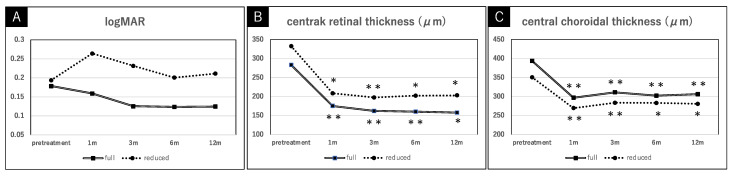
The serial changes in the logMAR (**A**), the central retinal thickness (**B**), and the central choroidal thickness (**C**) mean during the 1-year follow-up after the combination therapy, divided by treatment conditions. Under each treatment condition, there were statistical differences in the central retinal thickness and the central choroidal thickness at each time period, compared to before treatment. Data are expressed as mean and * *p* < 0.05, ** *p* < 0.01.

**Figure 3 pharmaceuticals-15-00483-f003:**
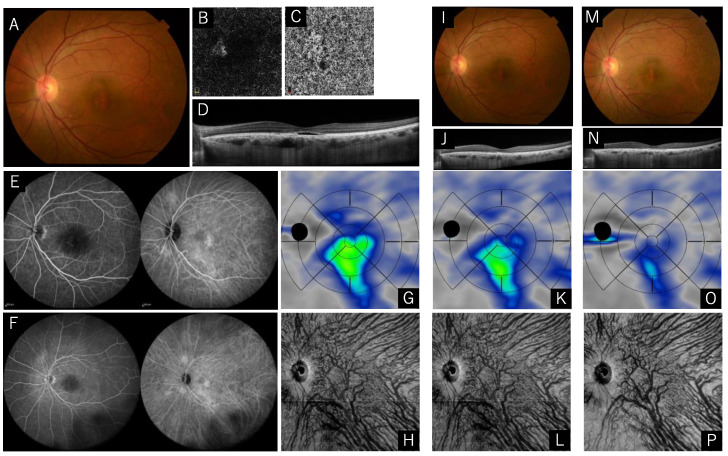
Images of the right eye of a 58-year-old male with pachychoroid neovasculopathy. (**A**–**H**) Images before treatment, and (**I**–**L**) three months and (**M**–**P**) one year after combination therapy. (**A**) Color fundus photograph shows the subretinal detachment. (**B**,**C**) Optical coherence tomography (OCT) angiography of the outer retina and the choriocapillaris slabs shows choroidal neovascularization. (**D**) The OCT horizontal scan thorough the central fovea shows subretinal fluid and pachyvessels with inner choriocapillaris attenuation. (**E**,**F**) Fluorescein angiography demonstrates faint leak points, and indocyanine green angiography presents hyperfluorescein spots and choroidal vascular hyperpermeability. (**G**) Choroidal thickness map shows greater choroidal thickness in central and inferior areas. (**H**) En face OCT image in the deep layer of the choroid represents dilated choroidal vessels, and horizontal watershed zone is not observed. (**I**,**J**) Three months after the treatment, subretinal fluid disappeared, and pachyvessels diminished. (**K**,**L**) Choroidal thickness map shows reduction in choroidal thickness in the central area, and the en face OCT image shows persistently dilated choroidal vessels and lost horizontal watershed zone. (**M**,**N**) No recurrence was exhibited at one year after the treatment. (**O**) Choroidal thickness map shows reduced choroidal thickness in the central area and the inferior area. (**P**) The constriction of choroidal vessels imaged by en face OCT persisted in the macular area, while the diminished choroidal vessels were also seen in the inferior area.

**Figure 4 pharmaceuticals-15-00483-f004:**
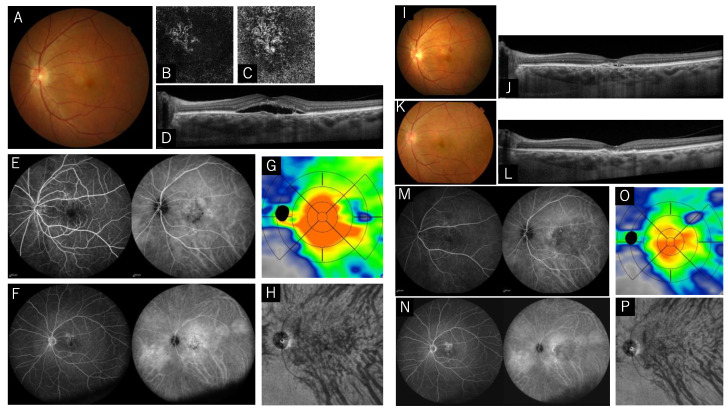
Images of the left eye of a 72-year-old male with pachychoroid neovasculopathy. (**A**–**H**) Images before treatment, (**I**,**J**) five months after combination therapy, and (**K**–**P**) one year after combination therapy. (**A**) Color fundus photograph shows the subretinal detachment in the macula. (**B**,**C**) Optical coherence tomography (OCT) angiography of outer retina and choriocapillaris slabs shows choroidal neovascularization. (**D**) OCT horizontal scan through central fovea shows subretinal fluid, retinal pigment epithelium undulation, and pachyvessels with inner choriocapillaris attenuation. (**E**,**F**) Fluorescein angiography shows diffuse leakage, and indocyanine green angiography (ICGA) shows multiple hyperfluorescein spots and wide multiple choroidal vascular hyperpermeability. (**G**) Choroidal thickness map shows thickened/thicker choroidal thickness, especially in the central area. (**H**) En face OCT image in the deep layer of the choroid shows dilated choroidal vessels, and horizontal watershed zone cannot be observed/or is unclear. (**I**,**J**) Five months after combination therapy, retinal hemorrhage was observed and shallow subretinal fluid appeared. (**K**,**L**) One year after combination therapy, orange-red raised lesion was observed in superior macula, with no exudative changes. (**M**,**N**) ICGA shows dark spot in macula and new polyps, and the condition was diagnosed as polypoidal choroidal vasculopathy. (**O**) Choroidal thickness map shows reduction in choroidal thickness in the central area. (**P**) En face OCT image shows constriction of choroidal blood vessels.

**Table 1 pharmaceuticals-15-00483-t001:** Clinical characteristics of patients at baseline.

	All (*n* = 29)	Full-Treatment Group (*n* = 16)	Reduced-Treatment Group (*n* = 13)	*p*-Value
age (years, (SD))	68.3 (8.5)	68.3 (10.2)	68.5 (5.7)	0.510 ^1^
sex Male/Female	25/4	14/2	11/2	1.000 ^2^
logMAR (average, (SD))	0.19 (0.29)	0.18 (0.25)	0.19 (0.34)	0.774 ^1^
central retinal thickness (μm)	305.2 (104.8)	283.1 (80.7)	332.4 (123.0)	0.245 ^1^
central choroidal thickness (μm)	374.4 (73.2)	393.8 (67.1)	350.7 (73.5)	0.203 ^1^
past treatment history (%, (*n*))	31.0 (9)	43.8 (7)	13.3 (2)	0.130 ^2^
diabetes mellitus (%, (*n*))	10.3 (3)	6.3 (1)	15.4 (2)	0.573 ^2^
hypertension (%, (*n*))	44.8 (13)	43.8 (7)	46.2 (6)	1.000 ^2^
smoking history (%, (*n*))	65.5 (19)	75.0 (12)	53.8 (7)	0.190 ^2^
spot size (μm)	4703.4 (1202.7)	4656.3 (1033.8)	4761.5 (1380.4)	0.982 ^1^

^1^: Mann–Whitney U test. ^2^: Fisher’s exact test. Age, logMAR, central retinal thickness, central choroidal thickness, and spot size are written as averages (standard deviations).

**Table 2 pharmaceuticals-15-00483-t002:** No-recurrence rate and the total number of additional treatments during the 1-year follow-up.

	All (*n* = 29)	Full-Treatment Group (*n* = 16)		Reduced-Treatment Group (*n* = 13)	
no recurrence (%, (*n*))	75.9 (22)	68.8 (11)		84.6 (11)	
anti-VEGF	0.28 (0.58)	0.31 (0.58)		0.23 (0.58)	
past treatment history	+ (*n* = 9)	+ (*n* = 7)	− (*n* = 9)	+ (*n* = 2)	− (*n* = 11)
no recurrence (%, (*n*))	66.7 (6)	57.1 (4)	77.8 (7)	100.0 (2)	81.8 (9)
anti-VEGF	0.22 (0.42)	0.29 (0.45)	0.33 (0.67)	0 (0)	0.27 (0.62)

Data in the anti-VEGF row are expressed as averages (standard deviation). VEGF, vascular endothelial growth factor.

**Table 3 pharmaceuticals-15-00483-t003:** No recurrence rate during the 1 year follow up divided into anti-VEGF agents.

	All (*n* = 29)	Ranibizumab Group (*n* = 24)		Aflibercept Group (*n* = 5)	
no recurrence (%, (*n*))	75.9 (22)	79.2 (19)		60.0 (3)	
past treatment history	+ (*n* = 9)	+ (*n* = 7)	− (*n* = 17)	+ (*n* = 2)	− (*n* = 3)
no recurrence (%, (*n*))	66.7 (6)	71.5 (5)	82.4 (14)	50.0 (1)	66.7 (2)

## Data Availability

Data are available within the article.

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
