# Peer review of "One-Year Outcome of Combination Therapy with Full or Reduced Photodynamic Therapy and One Anti-Vascular Endothelial Growth Factor in Pachychoroid Neovasculopathy"

_pharmaceuticals, 2022, doi:10.3390/ph15040483_

Round 1
Reviewer 1 Report
- Pachychoroid neovasculopathy (PNV) is a poorly defined term. As shown in Figure 3 G, the choroid does not seem to be ‘pachy’ at all. Please state the exact diagnosis of the eyes included in this study, like CSCR, PCV etc.
- In the two case presentations, please clarify the diagnosis in the text.
- Figures 3 and 4, which slab was used to generate the en face OCT structural images? Please also add OCTA en face images to show the neovascularization.
- Was CCT measured in the total 12X12 mm scan or the central macula? The author stated a fully automated choroid quantification algorithm developed by ZEISS. Has this algorithm been validated? Does this algorithm allow the visualization of the actual segemtation line of the choroid? How did the authors deal with the cases when the segmentation of the choroid was incorrect?
- Since some of the patients were injected with ranibizumab, while others were injected with aflibercept. The authors should take a close look into if there is any difference in the treatment outcome using these two drugs. Also, is combined therapy really necessary when using aflibercept?
Reviewer 2 Report
The authors evaluated a one-year treatment outcome after full or reduced PDT and anti-VEGF combination therapy for PNV. There are some limitations for the experimental design, for example, 2 different anti-VEGF agents (Eylea and Lucentis) were used for the intravitreal injections and the sample amount may not be sufficient enough to support the conclusion. However, the authors already given certain discussions in the manuscript and I will not provide extra comments on these. But I do have one question regarding Figure 2. In figure 2, it seems that the authors only did statistical analysis before and after the treatment. I am wondering whether there is any statistic difference between the full and reduced PDT treatment at each treatment time point. If possible, please add this information in the manuscript
Round 2
Reviewer 1 Report
The authors have addressed my concern and questions.
Reviewer 2 Report
I don't have further comments.